# Temporal and Spatial Variability of Volatile Organic Compounds in the Forest Atmosphere

**DOI:** 10.3390/ijerph16244915

**Published:** 2019-12-05

**Authors:** Francesco Meneguzzo, Lorenzo Albanese, Giorgio Bartolini, Federica Zabini

**Affiliations:** 1Institute for Bioeconomy, National Research Council, 10 Via Madonna del Piano, I-50019 Sesto Fiorentino (FI), Italy; lorenzo.albanese@ibe.cnr.it; 2“Fiorenzo Gei” Scientific Committee, Italian Alpine Club, 2 Via del Mezzetta, I-50135 Firenze (FI), Italy; 3Laboratory of Monitoring and Environmental Modelling for the Sustainable Development (LaMMA Consortium), Via Madonna del Piano 10, 50019 Sesto Fiorentino, Italy; bartolini@lamma.rete.toscana.it

**Keywords:** bioactive compounds, forest air, forest bathing, forest therapy, hiking trails, human health, monoterpenes, stress, volatile organic compounds

## Abstract

The healing effects of the forest are increasingly being valued for their contribution to human psychological and physiological health, motivating further advances aimed at improving knowledge of relevant forest resources. Biogenic volatile organic compounds, emitted by the plants and accumulating in the forest atmosphere, are essential contributors to the healing effects of the forest, and represent the focus of this study. Using a photoionization detector, we investigated the high frequency variability, in time and space, of the concentration of total volatile organic compounds on a hilly site as well as along forest paths and long hiking trails in the Italian northern Apennines. The scale of concentration variability was found to be comparable to absolute concentration levels within time scales of less than one hour and spatial scales of several hundred meters. During daylight hours, on clear and calm days, the concentration peaked from noon to early afternoon, followed by early morning, with the lowest levels in the late afternoon. These results were related to meteorological variables including the atmospheric vertical stability profile. Moreover, preliminary evidence pointed to higher concentrations of volatile organic compounds in forests dominated by conifer trees in comparison to pure beech forests.

## 1. Introduction

In recent years, attention has continued to grow with respect to the use of the forest environment for health purposes [1]. Plenty of recent evidence supports the value attributed to forest bathing, or forest therapy, derived from the Japanese *Shinrin-Yoku*, a kind of forest recreation traditionally practiced for improving physical and mental health and as a remedy for stress [2].

Significant positive effects on mental health, especially in people with depressive tendencies, were demonstrated by psychological analyses and physiological measurements, showing remarkable improvements in mood and depressive symptoms [3,4,5], a decrease in anxiety and stress levels [6,7,8], and a decreased risk of psychosocial stress-related diseases [9,10,11]. On the physiological side, the forest environment showed anti-hypertensive effects [10,12,13], promoted the regularization of the heart rate variability [14,15,16], increased the numerosity and activity of human natural killer cells [17,18,19], and induced the decrease of the blood glucose levels in diabetic patients [20].

Based on the above-mentioned health effects, forest therapy has been attributed a specific role in medical prevention. Healthcare programs based on forest therapy have been developed in some countries [21] and have recently been recommended on a large scale, with a sound economic foundation [22].

Among several factors able to trigger physiological activation in the forest environment, the inhalation of aromatic volatile substances, i.e., biogenic volatile organic compounds (BVOCs), plays an important role [17]. BVOCs, as low-molecular-weight compounds [23],, are emitted by plant parts above and below the ground, as well as by leaf litter, soil microbes, and insects, and show important functions for protection, defense, and communication among plants as well as between plants and other organisms [24]. All plant organs from flowers to roots can generate and release BVOCs [25], with leaves generally producing the highest emission rates [26].

Volatile terpenes represent most of the BVOC emissions in the forest environment, such as isoprene (which accounts for about 50% of the global emitted BVOCs per year), and monoterpenes (MTs, consisting of two isoprene units and accounting for about 15%) [27]. Thus, isoprene and MTs represent the major components of forest aerosols [26].

Based on decades of studies, reliable catalogues of the BVOC emission potential of leaves and needles of major European forest tree species are nowadays available. Notably, trees able to store MTs, such as conifer species, are not necessarily the highest potential emitters, with deciduous trees typical of mountainous forests in European temperate areas, such as beech (*Fagus sylvatica*) and chestnut (*Castanea sativa*), potentially greatly exceeding widespread conifer species such as Norway spruce (*Picea abies*) and silver fir (*Abies alba*) [28]. However, some studies showed that conifers can be important emitters of their respective typical MTs, among which *α*-pinene and *d*-limonene often stand out, which are endowed with recognized and important biological activities [29].

Beyond the large differences among forest tree species, the study of the variability of BVOC emission rates is hindered by their high reactivity with other compounds in the atmosphere, and by the high variability of biosynthesis and emission mechanisms, leading to remarkable intraspecific differences in terpene emission with regard to both total emissions and chemical profile [30]. Emission rates from leaves depend on several factors, such as physiological (age and developmental stage of plant) and physicochemical aspects (i.e., related to temperature, stomatal openness, and leaf structure) [31].

Temperature and light (radiation) were shown to be key environmental mechanisms for the synthesis and emission of BVOCs. Generally, isoprene emission is high dependent on light as well as on temperature due to its high volatility [31]. MT emission may show different sensitivity to light and temperature, also depending on the presence or absence of storage structures [32]. In plants with BVOC storage compartments, the emissions are basically temperature-dependent because temperature is able to volatilize compounds from stored pools, whereas in plants not storing BVOCs the emission rate is light and temperature-dependent [33,34]. Temperature variability affects also the VOC fluxes from the forest floor [35].

A recent study performed by Korea’s National Center for Forest Therapy focused on the variability of BVOCs concentration in the atmosphere of a *Pinus densiflora* forest, on a fixed site, during daylight hours (one-hour measurements starting at 08:00 am, 12:00 pm, and 17:00 pm), and once a month for a year [36]. The study was motivated by the consideration of BVOCs as an important resource in forest therapy due to their wide-spectrum healing effects, and was carried out by means of absorbent tubes and subsequent desorption and gas chromatographic analysis.

A remarkable seasonal variability was found, with the concentration of total BVOCs being highest in summer, followed by spring and fall, and lowest in winter. In summer, the relative concentration of *α*-pinene was also the highest, showing that not only total BVOCs but also individual components may change seasonally. On a daily basis, BVOC concentrations were found to increase from 08:00–09:00 am to 12:00–13:00 pm, and again, marginally, up to 17:00 pm, in contrast to previous studies which found higher concentrations at sunrise and sunset. However, admittedly, the discrete sampling prevented drawing definitive conclusions.

This study presents the results of a few high-frequency monitoring campaigns of total volatile organic compound (TVOC) concentrations both on fixed sites and along paths (a forest path and few hiking trails) far from anthropogenic sources, with each campaign carried out for several hours during daylight. Beyond the representativeness of the measurements of TVOCs with regard to BVOCs, such relative remoteness was deemed to minimize the reaction of terpenes with anthropogenic pollutants that, in contaminated environments such as urban forests, might enhance photochemical pollution [30], possibly offsetting the BVOC-related health benefits. No other previous studies investigated the variability of the concentration of TVOCs or BVOCs, in the forest air with such temporal and spatial detail, let alone along long paths (tens of km), spanning different elevations, climates, and forest compositions.

## 2. Materials and Methods

### 2.1. Measurement of Volatile Organic Compounds

TVOCs in the forest atmosphere were measured by means of a portable (0.72 kg) photoionization detector (PID, model Tiger VOC detector, Ion Science Ltd., Fowlmere, Royston, UK), with detection limits from 0.001 ppm (1 ppb) to 20,000 ppb. The PID was equipped with a pump, aspiring the ambient air at a rate of 220 mL/min, and an ultraviolet, 10.6 eV lamp allowing the ionization of organic substances in the aspired air, including all the MTs found in BVOCs. The resulting electric current flowing between two electrodes was measured and amplified, and transformed into a concentration level of the ionized substance or group of substances.

Factory calibration was performed at the temperature of 21.5 ± 0.1 °C and atmospheric pressure with respect to the substance isobutylene, with an estimated measurement uncertainty of ±2%. The concentration of other substances was derived after the application of a response factor that, for TVOC, was equal to 1. While practically immediate, this detection method did not allow the identification of single compounds in TVOCs, which would require a gas chromatograph. This issue is discussed further in Section 4.

### 2.2. Study Sites and Weather Stations

Figure 1a shows the location of the study sites in a wide geographical context, across Tuscany and the Emilia-Romagna regions, Italy, labeled as S1 (hilly site), S2 (forest path), and H1 and H2 (hiking trails). The considered weather stations (W1 to W9) are represented too. The study areas are shown in detail in Figure 1b–e, with the forest path (S2) shown in yellow, and the hiking trails (H1 and H2) represented by red and blue lines.

Table 1 shows the list of the study sites and the weather stations, along with the respective geographical coordinates (for weather stations), elevation, and data collected.

All weather data were collected every 15 min and consisted of the respective averages over the previous 15 min. The weather stations W1 to W8 belonged to the monitoring network of the Regional Functional Center for Weather—Hydrological Monitoring, Tuscany, Italy (http://cfr.toscana.it/). Weather station W9 belonged to the monitoring network of the “Reggio Emilia Meteo” association (http://www.reggioemiliameteo.it/), and the respective data (temperature and wind) were retrieved from the public web site.

### 2.3. TVOC, Geographical, and Meteorological Data: Measurement, Logging and Merging

The measurement sessions on the fixed site S1 were carried out with PID pump placed 1.2 ± 0.1 m above the ground, with such height corresponding approximately to the height of the nose of a 1.75-m tall person sitting on a chair. The measurement sessions along pathways (S2, H1, and H2) were carried out with the PID fixed on the backpack. The pump was again placed 1.2 ± 0.1 m above the ground, and the paths were always traveled on foot, in order to avoid any contamination from engine exhaust gases. However, most of the routes could not be traveled by other means.

The PID logged data every 2 s, with each data string consisting of the time of measurement and TVOC concentration level in ppm unit (mg/kg). For each measurement session, the lowest detected concentration (which could be positive or negative, due to the factory calibration used), was subtracted from all the other data, in order to remove, as much as possible, background TVOC concentration level. Such background level likely included BVOCs too, and thus resulting TVOC concentration levels could be considered as lower estimates of the actual BVOC concentration levels, especially due to the remoteness of the considered sites from any anthropogenic source.

While in principle changes of the levels of anthropogenic VOCs during time spans of a few hours could not be ruled out, for example due to changes in the atmospheric vertical stability or to changes in the emission rates from anthropogenic sources, the above-mentioned remoteness should keep such uncertainty at comparatively low levels. This topic is further discussed in Section 4. Finally, since this study aimed at a preliminary assessment of the relative variability in time and space of BVOC concentration levels within the forest atmosphere, the accurate assessment of the respective absolute concentration levels remains outside the scope.

Limited to the measurements along pathways, continuous geographic localization was performed by means of Global Positioning System (GPS) software installed in a commercial smartphone (Open GPX Tracker, https://apps.apple.com/it/app/open-gpx-tracker/id984503772), with data logging every 7 s, including the time of measurement, latitude, and longitude. The height above sea level (a.s.l.) was assigned to each geographical point, based on the dataset of the National Aeronautics and Space Administration, Shuttle Radar Topography Mission (NASA’s SRTM), 30-m horizontal resolution, by means of an online conversion software (GPS Visualizer, https://www.gpsvisualizer.com/convert_input). The PID and the smartphone were synchronized, and then TVOC and geographical data were merged based on the respective measurement times, with TVOC data interpolated to the time points of the GPS data.

In order to assign to each TVOC data point, at any time, a complete set of meteorological data (temperature, global radiation, and wind intensity), the data collected at the weather stations were interpolated as follows. At local elevations below or above the range of elevations of the considered stations, the air temperature measured at the nearest weather station was interpolated based on the local elevation, according to the standard vertical gradient in the lower troposphere of −0.6 °C/100 m [37]. Otherwise, a linear interpolation to the local elevation was performed, based on data collected at the weather stations located at the closest elevations below and above the local one.

The wind intensity was linearly interpolated to the local elevation, based on two of the weather stations located at elevations closest the local one. Global radiation data were available only at a single weather station for each study site. However, due to the observed low-level cloudiness around the ridge along the H1 hiking trail, the global radiation data at elevations above 1600 m a.s.l. were corrected to random levels between 0 and 100 Wm^−2^—a range representative of cloudy conditions—whenever the relative humidity at the weather station W5 was at the level of 95% or above. While this was a somehow arbitrary choice, adopted for representation purposes, the correction was imposed by the relative remoteness of the weather station W5 (measuring the global radiation), its location far below the mountain ridge, and the local character of the cloudiness over the ridge itself. Moreover, knowing the exact level of the radiation below 100 Wm^−2^ was practically useless for the purposes of this study.

## 3. Results

### 3.1. Hilly Site

#### 3.1.1. Clear and Calm Days

Figure 2a–c shows TVOC concentration and weather data series for the hilly site, labeled as site S1 in Table 1, during days with generally clear skies, low wind (average wind intensity over 15 min lower than 3 m/s), and relatively high temperatures. TVOC concentration, temperature, and global radiation are shown in Figure 2a–c, respectively.

The relative changes of TVOC concentration, during any given day, spanned from 0.5 to more than 0.9 ppm, i.e., around the magnitude of the highest concentration levels commonly observed in the forest air [29]. During any given day, TVOC relative concentration appeared to peak during the time period 13:00–14:00 pm to 16:00 pm solar time, seemingly following the peak air temperature which lags behind peak radiation rather than global radiation itself. However, dependence of TVOC concentration on radiation could not be ruled out, for example due to the accumulation and persistence of MTs in the lower atmosphere. Forest in site S1 was locally dominated by both evergreen trees (storing MTs) such as cypress and eucalyptus and deciduous trees such as oak and poplar, suggesting dependence on both temperature and radiation [32]. However, looking at different days, no evidence arose with respect to a monotonic relationship between the maximum amplitude of the daily concentration cycle and the peak temperature.

Radiation played a more remarkable role in late afternoon (approximately, after 16:00 pm), when air temperature decreased slowly and was still relatively higher than in the morning, while TVOC concentration dropped much faster than the respective rate of increase observed in the morning. Likely, emissions from deciduous trees dropped quickly with the vanishing radiation. This could be confirmed based on the event that occurred on 1 September 2019, when the sudden drop in TVOC concentration, starting around 15:30 pm, followed the fast decrease of global radiation (related to a sudden storm approaching the site), with a short delay (about 30 min). The air temperature dropped too, down to approximately the levels shown in the same time period during the days of 28 September and 6 October, when TVOC concentrations did not drop significantly.

Another important factor contributing to the falling concentration in late afternoon might be the effect of the lowering of radiation levels on the atmospheric vertical stability, especially during clear and calm days, and thus on the diffusivity of aerial substances down to the ground. This topic will be discussed further in Section 4.

Finally, on 28 September 2019, the initial TVOC concentrations (around 07:00 am) were relatively higher than subsequent levels, until about 10:00 am, despite lower radiation and air temperature. This behavior could be suggestive of some relationship between TVOC concentrations and other atmospheric properties, such as the atmospheric vertical stability, which will be discussed further in Section 4.

#### 3.1.2. Other Days

Figure 3a–d shows TVOC concentration and weather data series for the hilly site, labeled as site S1 in Table 1, during days with generally cloudy skies, lower temperatures, or relatively sustained wind. TVOC concentration, temperature, global radiation, and wind intensity are shown in Figure 3a–d, respectively.

The relative changes of TVOC concentration, during any given day, did not exceed 0.5 ppm, i.e., substantially less than on clear and calm days. The global radiation was not much lower than in the clear days, although generally more irregular. The air temperature was generally lower, which—along with frequent drops in radiation—could explain the generally lower levels of the amplitude of TVOC concentration cycles.

On 24 August 2019, the delayed peak in the air temperature, around 16:00 pm, was associated with the peak in TVOC concentration, which could have been favored also by the decreasing wind intensity and the relatively high radiation levels. On 9 September, the sudden radiation drops after about 13:00 pm and the simultaneous increase in the wind intensity could have been the drivers of TVOC concentration decline after a weak peak around 10:00 am, when the wind was low and the radiation already sufficiently high.

Over two days, 9 September and 15 September, with measurements beginning early enough in the morning when there was sufficient radiation (contrary to other days) and the wind was low, TVOC concentration was higher in the early morning (07:00–08:00 am) than in the following few hours. Again, this evidence possibly points to a relationship between TVOC concentration and the vertical atmospheric stability, which was already noted in Section 3.1.1 and will be discussed further in Section 4.

On the other hand, TVOC concentrations appeared to fall generally less steeply in the afternoon than in the clear and calm days considered in Section 3.1.1. Since the solar radiation affects the atmospheric vertical stability more substantially during clear and calm days, this behavior could be again suggestive of an important role played by the atmospheric vertical stability. This topic will be discussed further in Section 4.

### 3.2. Forest Path

Figure 4a–d shows TVOC concentration and weather data series for the forest path, labeled as site S2 in Table 1, measured on 11 September 2019. Figure 4e–h shows the same quantities for site S2, measured on 5 October 2019.

Based on weather data, shown in Figure 4b–d, 11 September 2019 was a clear and calm day, with very low wind; air temperatures were in the range 16 to 23 °C. The cyclical trend of TVOC concentration, with local peaks around the lowest elevation of the forest path, and local minimums at the highest part of the path, points to a remarkable spatial variability, with a maximum amplitude around 0.4 ppm. Such variability could be related to either the air temperature (decreasing with the elevation at any fixed time), or the plant species and status, soil emissions, and so on.

The above-mentioned peaks occurred at very different levels. In particular, TVOC concentration peaks were substantially higher in the early morning (before 09:00 am) and just after 12:00 pm, than around 10:00 am, 14:30 pm, and 16:00 pm. Similar to the behavior observed in Section 3.1, temperature could have driven up TVOC emissions and concentrations during the hottest few hours of the day, while the radiation-driven atmospheric vertical stability could have affected TVOC concentration levels near the ground in early morning (increase) and late afternoon (decrease).

Based on weather data, shown in Figure 4f–h, 5 October 2019, was a calm day with very low wind, but with alternation of cloudiness and sunny spells, as apparent in the radiation data. The air temperatures ranged from about 13 to 17 °C and no rain was observed during the day.

A cyclical trend of TVOC concentration appeared again in Figure 4e, qualitatively very similar to that observed in Figure 4a, with local peaks around the lowest elevation of the forest path, local minimums at the highest part of the path, and maximum amplitude around 0.55 ppm. Such greater amplitude could have been due to the wild changes in radiation.

The absolute peak in TVOC concentration was again observed in early afternoon, in particular between 12:40 and 14:00 pm, during a prolonged stop at the lowest site of the route. Interestingly, TVOC concentration increased by more than 0.25 ppm since the arrival at the lowest site, possibly following the sudden spike in radiation, with air temperatures around the peak of day. The second peak in radiation occurred around the highest elevation site of the route (around 14:30 pm), and could have helped to retain a relatively high concentration level. The subsequent fast drop in radiation could explain the substantial absence of further peaks at the lowest site.

### 3.3. Hiking Trails

Figure 5a–e shows TVOC concentration (both superimposed on a topographic map and as a chart) and weather data series for the uphill part of the hiking trail labeled as H1 in Table 1, as measured on 12 October 2019. The lowest TVOC concentration level in Figure 5b is above zero because the offset was computed over the whole day, i.e., over the entire hiking trail, including the downhill part, which is described below. The first part of the route, labeled as A in Figure 5b, coincided with the forest path (site S2 in Table 1) analyzed in Section 3.2.

TVOC concentration increased by nearly 0.6 ppm along part A of the route (forest path, site S2 in Table 1, with dominance of chestnut, ash, black pine, and silver fir trees), approximately from 09:00 to 09:45 am, which occurred with low wind, relatively high air temperatures, and moderate radiation. This situation could have mimicked early morning conditions in summer.

Along part B of the route, steeply rising from about 1150 to 1450 m a.s.l., and mostly dominated by beech and silver fir trees, TVOC concentration remained at comparatively high levels. From approximately 09:45 to 10:30 am, the global radiation was still moderate, the air temperature decreased by about 3 °C, and the wind intensity was still low, similarly to part A of the route. The subsequent small local peak in TVOC concentration, along part C of the route, occurred along with a local minimum of both radiation and air temperature, possibly reinforcing the hypothesis that early morning-like vertical atmospheric stability could favor relatively high TVOC concentrations near the ground. This hypothesis will be discussed further in Section 4. In addition, the change in the composition of the forest vegetation, with beech becoming dominant over fir trees, could have played a role.

Along parts D and E of the route, with elevation eventually reaching about 1600 m a.s.l. and the trail emerging outside a pure beech forest and reaching a mountain grassland, TVOC concentration gradually dropped to a minimum. During these parts of the route (from around 10:45 to 11:45 am), the wind intensity increased as the mountain ridge was approached, the radiation reached the daily peak (over 600 Wm^−2^), and the air temperature changed slightly. Although the relatively high radiation levels should have favored the BVOC emissions from beeches, the neutral to unstable vertical atmospheric profile could have diluted the VOCs across a deeper atmospheric layer, thus lowering TVOC concentrations near the ground. Moreover, the change from mixed beech–fir tree forest to pure beech forest could have contributed to the observed drop under the specific meteorological conditions.

The absolute peak in TVOC concentration, observed along part F of the route (from around 12:00 to 12:45 pm), is harder to explain. Most of the route crossed a mountain grassland, only occasionally reentering the forest; air temperature did not change remarkably, while radiation oscillated wildly between nearly zero (ridge inside low clouds) and almost 500 Wm^−2^. Notably, the southwesterly wind intensity increased up to the daily maximum of 5 ms^−1^, which might suggest the advection of BVOCs from the nearby forest and the respective accumulation near the ground on the upwind side of the ridge. The same factors could also explain why TVOC concentration dropped by only about 0.3 ppm, well below what one could have expected, along part G of the route (12:45 to 13:45 pm), entirely developing around the ridge and on mountain grasslands, with very low radiation and air temperatures at the daily minimum (around 7 °C).

Figure 6a–e shows TVOC concentration (both superimposed on a topographic map and represented as a chart along with the local elevation) and weather data series for the downhill part of the hiking trail labeled as H1 in Table 1, again measured on 12 October 2019. The latter part of this route was slightly different from the uphill path, while eventually getting to the same site, and is labeled with a different letter (H).

TVOC concentrations along most of the downhill route were much lower than along the uphill route, which is apparent after comparing the maps in Figure 5a and Figure 6a. This comes as little surprise because air temperatures were generally lower and global radiation even more so due to increased cloudiness and the progression of time in the afternoon. The lower part of the downhill route (parts E, D+C, and H in Figure 6b) could have been deeply affected by the changed conditions, with TVOC concentration levels lower by 0.25 to 0.45 ppm in comparison with the uphill route.

The absolute minimum in TVOC concentration levels in part F of the downhill route shown in Figure 6b and corresponding to the red colored tags in Figure 6a is especially suggestive. Such levels were as much as nearly 0.9 ppm lower than the levels observed along the same part of the route just 3 h earlier. Indeed, this part crossed mainly mountain grasslands, and most of observed VOCs were likely to be the result of the advection from the nearby forest. With lower radiation and somewhat lower wind intensity, likely both BVOC emissions from forest trees and the respective advection decreased, resulting in remarkably lower TVOC concentrations.

Figure 7a–e shows TVOC concentration, both superimposed on a topographic map and represented as a chart along with the local elevation, and weather data series for the ring-shaped hiking trail labeled as H2 in Table 1, measured on 29 September 2019.

TVOC concentration levels oscillated with a maximum amplitude greater than 0.75 ppm along the ring-shaped route. The highest peak occurred along the part of the route labeled as A in Figure 7b, in early morning (08:00 to 08:40 am) at elevations of 1400 to 1550 m a.s.l., in a mixed beech–fir trees forest with both silver firs and Douglas firs. Weather conditions were characterized by moderate global radiation (300–400 Wm^−2^), relatively low temperature (9–11 °C), and wind intensity (3 to 4.5 ms^−1^), likely leading to a shallow neutral to unstable vertical atmospheric profile.

Subsequently, part B of the route (08:45 to 10:15 am) developed across mountain grasslands at higher elevations (from more than 1800 to about 1750 m a.s.l.), with lower temperatures (about 8 °C), lower radiation levels (200–300 Wm^−2^), and stronger wind (up to 5.5 ms^−1^). TVOC concentration levels dropped to a prolonged absolute minimum, starting from the ridge and further along the leeward side (the wind was blowing from the south-west). Likely, no effective advection of VOCs occurred from forests located at least 200 m below the ridge, and constituted (on the windward side) by pure beech stands.

Along part C (10:15 to 11:30 am) at elevations between 1650 and 1750 m a.s.l., alternately crossing grasslands and forest stands with both beech and fir trees, TVOC concentrations gradually increased, eventually by nearly as much as 0.6 ppm, possibly in response to both the presence of tree vegetation and increased radiation, with the latter reaching about 500 Wm^−2^. At about 11:30 am–12:30 pm (part D), TVOC concentration measurements were taken at a fixed point (“C. Battisti” refuge, managed by the Italian Alpine Club) at the elevation of 1765 m a.s.l., immersed in a mixed beech–fir tree forest. TVOC concentrations initially increased by further 0.1 ppm, later dropping steeply by as much as 0.45 ppm. The reason of such decline in TVOC concentration may have been the trend in radiation, initially increasing up to more than 700 Wm^−2^ and later dropping to an average of 350 Wm^−2^.

During part E of the route (12:30 to 12:50 pm), developed on the ridge covered by a grassland at elevations around 1780 m a.s.l., TVOC concentrations remained stationary around a local minimum. Later, they increased remarkably by about 0.25 ppm at the local peak during the descent across a historic forest (“Abetina Reale”) dominated by silver fir trees, along with beeches and Norway spruce (Part F, 12:50 to 13:40 pm), down to about 1400 m a.s.l. (location of the “Segheria” refuge, managed by the Tuscan-Emilian Apennine National Park). While slightly stronger radiation (up to about 440 Wm^−2^) and relatively higher temperature (up to about 12 °C) could not be ruled out, the forest vegetation was likely the main driver for local BVOC emissions and, consequently, increased concentrations near the ground.

Finally, a forest path crossed a pure beech forest in parts G and H of the route (ascent to the ridge at the elevation of 1570 m a.s.l., and descent to the starting point of the trail, respectively). Decreasing radiation levels or the forest composition, or both, could have contributed to relatively lower TVOC concentrations (about 0.2 ppm below the local peak observed in the “Abetina Reale” forest).

## 4. Discussion

The meaningfulness of this study and the related findings relies upon the representativeness of TVOC concentrations measured by means of a PID with respect to the biogenic components (BVOC). Since the components of the detected TVOCs were not analyzed in any study sites and paths, in principle their origin was unknown, i.e., they could be either biogenic or anthropogenic, or both. However, the remoteness of the areas into consideration from anthropogenic sources, as well as the adoption of a variable offset concentration level equal to the absolute minimum during any measurement day, with respect to which all the other data were referred, should have minimized the chance of substantial contributions of anthropogenic sources to reported TVOC concentration data. Based on the above, in the following, unless explicitly specified, the volatile organic compounds under consideration will be referred to as BVOCs.

The first and basic findings of this study concerned the remarkable variability of BVOC concentrations in time and space, both on fixed sites and along paths, as well as in different environments, i.e., low hill, mid-mountain sites and mountain sites up to over the upper limit of tree vegetation. Indeed, within the same day, relative changes of BVOC concentration commonly exceeded 0.5 ppm, often even reaching levels as high as 0.9 ppm, on the order of magnitude of the highest concentration levels observed in the forest air, i.e., around 1 ppm [29]. Such changes occurred within time scales as short as a few tens of minutes, and within distances of the order of several hundred meters along paths and hiking trails.

The takeaway message from these findings is that, as far as BVOCs are considered an essential resource contributing to the forest healing effects, forest therapists and forest bathers should carefully choose the right times and forest sites and paths even on clear and calm days. Hikers could incorporate this information into their walking plans in order to benefit from BVOC inhalation along with the advantages derived from the physical exercise.

Building a comprehensive quantitative model for BVOC concentration as a function of time of the day and meteorological data, even on a single site and for clear and calm days, would require much more extensive and continuous measurements. As well, the seasonality of emission rates from plants and soil should be included, in turn depending on the specific forest composition. The structure and the hourly evolution of the atmospheric boundary layer, including its vertical stratification, should be accurately represented too. The climatology of BVOC concentrations along a path would be even more challenging, due to changing forest composition and exposure of the path. On the other hand, mapping BVOC concentration near the ground is just in its infancy, and the road ahead is still very long.

However, some insight could be gained, limited to a single day and on a fixed site. Based on data collected on site S1 during 28 September 2019, shown in Figure 2a–e and endowed with the largest daily coverage, the series of 30-min average concentrations of volatile organic compounds was linearly regressed against air temperature and global radiation. Such regression was carried out both during the whole period of measurements, and separately for the morning (07:30 am to 12:30 pm) and afternoon (12:30 to 20:30 pm). The results are shown in Figure 8.

Apparently, the likely, subtle effects produced by the lowest radiation levels in early morning and late afternoon were represented much more accurately by the separate regressions performed for morning and afternoon. In addition, the peak concentration, observed around 15:00 pm, was better represented too, although marginally.

Equations (1) and (2) show the regression equations for morning and afternoon, respectively:TVOC = −1.14 + 0.09·T − 0.0005·R,(1)
TVOC = −2.26 + 0.11·T + 0.0005·R,(2)
where TVOC is concentration (ppm), T is temperature (°C), and R is radiation (Wm^−2^).

Equations (1) and (2) explained 82% and 97% of the variance, respectively, and were both significant at the 1% level, based on the F statistics. While the dependence of concentration on temperature was quite similar, the dependence on radiation was opposite, being negative in the morning and positive in the afternoon. Likely, this reflects the fact that temperature was the main forcing for emissions from plants [33,34], as well as from soil [35], while radiation drove the evolution of the atmospheric boundary layer [38]. The following is a preliminary explanation of these results.

In the morning, temperature rises rapidly, stimulating higher emissions, while the increase in radiation deepens the ground-based boundary layer, through which the emitted compounds are diluted, thus counteracting the effect of increasing emissions with regard to ground concentration. In early morning, the shallow boundary layer leads to relatively higher concentration despite the low emission levels. In the afternoon, declining radiation eventually determines a ground-based stable atmospheric layer, in turn surmounted by a residual mixing layer [38], thus hindering the accumulation of volatile organic compounds near the ground, despite relatively high emission rates driven by slowly decreasing temperature.

In an attempt to generalize the above-discussed findings, Figure 9 shows a simple conceptual model, applicable to clear and calm days.

Cloudiness prevents most of the solar radiation from reaching the ground and the tree foliage, primarily leading to reduced BVOC emissions, as well as to the limitation of atmospheric vertical mixing, with the latter hindering the ground accumulation of BVOCs early in the morning while attenuating the vertical dilution around noon in comparison with clear conditions. Overall, based on the results presented in Section 3, cloud cover seems to lead to lower BVOC concentrations near the ground. Strong winds drive remarkable horizontal advection of substances in the atmosphere, which can unpredictably lead to higher or lower BVOC local concentrations near the ground, depending on the presence, composition, and emission rates of forest stands on the windward side, as well as on other uncontrollable factors such as turbulence, leeward wakes, and other disturbances to the atmospheric vertical mixing.

Within the level of significance allowed by the measurement campaigns carried out on a limited sample of natural areas in central Italy, over a few days during less than three months between August and October in a single year, and referring to the only daylight hours, some interpretation and a few preliminary guidelines can be summarized as follows:Early in the morning, the soil heats up due to the emerging solar radiation, and a strong mixed layer develops and grows in the atmosphere just above the ground, leading to a shallow unstable boundary layer. Although BVOC emissions from trees are still limited by low levels of radiation and temperature, BVOCs diffuse within a limited vertical layer and can accumulate near the ground, up to relatively high concentration levels. Early morning could to be a relatively good time for the contribution of BVOCs to forest healing effects.Mid-morning, the atmospheric mixed layer grows rapidly in depth, driven by the increasing radiation, while BVOC emissions are still moderate. As a result, BVOCs diffuse within a relatively deep vertical layer and their accumulation near the ground can temporarily decrease in comparison to early morning. Mid-morning could be a relatively disadvantaged time.In the hours just after noon, with solar radiation and air temperature around their peaks, BVOC emissions peak too, leading to the highest concentrations near the ground despite the high dilution rate of BVOCs within the deep atmospheric mixed layer. The middle of the day, and early afternoon could be the best time for the contribution of BVOCs to forest healing effects, provided that other factors, such as those defining the bioclimatic comfort, are at acceptable levels.Late in the afternoon, with vanishing solar radiation, a stable atmospheric layer develops just above the ground, leading to a shallow stable layer with very limited vertical mixing, in turn surmounted by a residual mixing layer. The combination of declining BVOC emissions, due to low radiation levels and falling temperature, and the inhibition of their mixing from elevated sources (tree foliage) down to the ground, can lead to the lowest concentration levels. Late afternoon could be the most disadvantageous time for the contribution of BVOCs to forest healing effects.With regard to BVOC inhalation, during clear and calm days, a preliminary planning suggestion for hikers can be issued that forest trails should be preferably traveled in early morning and from noon to early afternoon, rather than in mid-morning and late afternoon. Depending on hiking miles, the latter periods could be used to rest at mountain refuges, or to walk through areas devoid of forest vegetation.Cloudy or windy days appear less suitable at any time to enjoy the BVOC-related forest healing effects due to generally lower and largely unpredictable concentration levels, adding this flaw to the generally lower level of comfort.Based on the results of this study, and under the specific meteorological conditions, conifer trees seemed to be more efficient with regard to BVOC concentrations in the forest air, at least in comparison with beech trees (the most widespread tree in Italy, concentrated in mountainous areas and especially in the Apennines, particularly at elevations above 1200 m). Thus, when different alternatives are available, pure fir or mixed beech-fir forests could be preferred over pure beech forests. However, the evidence was quite limited, and the above statement might not apply to different deciduous tree species, which are widespread at lower elevations.

The presented findings are partially contrasting with results recently obtained in Korea, where peak BVOC concentrations were found around 17:00 pm, and the lowest in the early morning [36]. However, different sampling frequencies, forest environments and latitudes seriously limit the meaningfulness of a comparison between the two studies. To the authors’ best knowledge, no other studies to date match the requirements for a meaningful comparison.

The advantages of using a PID for the measurement the concentrations of volatile organic compounds in forest air consist of the very high measurement and logging frequency (2 s), allowing to detect changes in real time on fixed sites, and about every 2 m along paths. The portability of the instrument, as well as the low cost of measurements represent further advantages. The results of this study demonstrate that high frequency measurements are needed in order to capture the substantial variability of BVOC concentrations in forest air, occurring within a few tents of minutes and several hundred meters.

The main and substantial disadvantage consists of the unselective measurements, which leave unknown the composition of the detected compounds and their safe attribution to plant emissions. The detection of the specific composition of volatile organic compounds, enabling their attribution to biogenic or anthropogenic sources and thus providing a more accurate representation of BVOC concentration, with an analysis of possible changes in time and space of the composition of BVOCs in the forest air, would require completely different devices. For example, absorbent tubes that must be deployed at least in triplicate at any point of measurement, require at least 60 min for any absorption session, and must be thermally desorbed offline [36].

The application of methods aimed at detecting the specific components of BVOCs, like the one mentioned above, would be practically unfeasible in the case of paths and hiking trails. However, a recommended direction for future research with respect to the characterization of forest sites and paths in terms of BVOC concentrations concerns the coupling of a high-frequency PID with analytical devices, such as absorbent tubes, in order to calibrate the PID-based measurements at selected sites.

This direction aims at defining the relative contribution of biogenic compounds to the total concentration of volatile organic compounds, and likely even more importantly the detailed composition of BVOCs, which could change with season, time of day, weather conditions, composition of the forest, and soil conditions [36]. Due to the remarkable specialization of different components of BVOCs with regard to their biological activities [39], the functionality, or healing effectivity, of a forest site or path would require the knowledge of the concentration not only of total BVOCs, but also of the relative abundance of their most active constituents, and the respective variability in time and space.

Additional recommendations for further research can be summarized as follows:Beyond the quantities considered in this study, other meteorological disturbances such as drought and accumulated rain in previous days should be taken into account.BVOC concentration daily cycles in different seasons, such as spring, early summer, late autumn, and winter, should be investigated, also aimed at building up comprehensive year-long models.Instead of data from sparse weather stations whose interpolation could lead to remarkable uncertainties, the use of high-resolution, gridded meteorological data over the geographical domain into consideration is recommended, for example extracted from reanalyses or very short-term forecasts generated by high-resolution mesoscale atmospheric models.A substantial amount of research is needed in order to score different tree types and forest compositions with regard to their effect on the variability of BVOC concentrations in the forest air, including the dependence on season and weather conditions, as well as accounting for emissions from both plants and soil.

This study should be considered as a starting point, and possibly as a methodological reference, for further investigation and research, following the above-listed recommendations. However, the basic finding and its consequences appear indisputable: as long as BVOCs are considered as an essential resource contributing to forest healing effects, detailed knowledge of the respective concentration patterns in forest air and careful planning of practices are needed in order to substantially optimize the benefits for human mental and physiological health.

## 5. Conclusions

Knowledge of the variability of BVOC concentrations in forest air could contribute to the optimization of forest healing effects. In particular, it could assist in planning forest therapy sessions and practices, and allow hikers to benefit more greatly from their walking along forest and mountain trails. For the first time, this study demonstrated that the variability scale of BVOC concentrations could be comparable to the absolute concentration levels within time scales of less than one hour and spatial scales of several hundred meters.

These results were achieved by using a photoionization detector that, while not allowing the detection of individual components, allowed high-frequency measurements both on fixed sites and along paths. The remoteness of the study areas from anthropogenic sources, along with suitable data processing, reasonably ensured the representativeness of the measurements of the total concentration of volatile organic compounds with regard to the biogenic ones.

Based on empirical evidence, limited quantitative relationships, and a conceptual model, provisional guidelines have been formulated about the evolution of BVOC concentrations during daylight hours, suggesting the likelihood of peak BVOC concentration during few hours after noon, a secondary peak in early morning, and lowest concentration in late afternoon during clear and calm days. In addition, a very preliminary conclusion was drafted that conifer trees are more efficient than beeches with regard to BVOC concentration in forest air.

## Figures and Tables

**Figure 1 ijerph-16-04915-f001:**
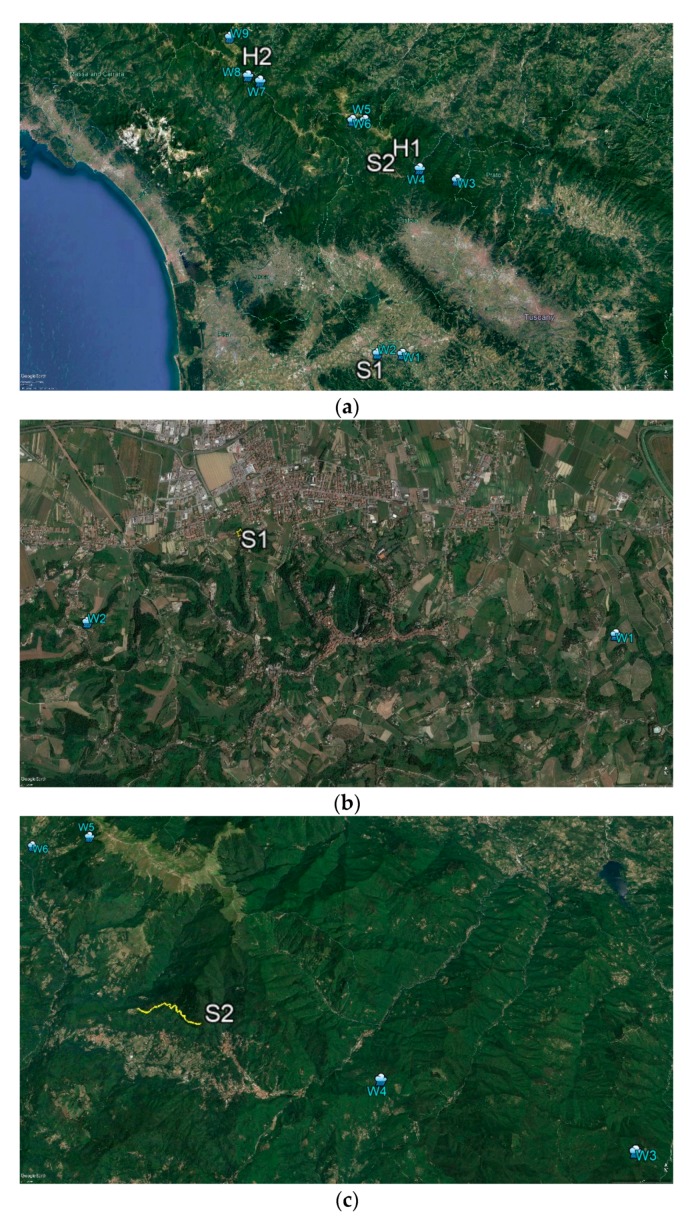
(**a**) Study sites (S1, S2, H1, and H2), and weather stations (W1 to W9), in a wide geographical context (central and northern Tuscany, Italy). (**b**) Detail over the (hilly) study site S1. (**c**) Detail over the study site S2, with the forest path represented by a yellow line. (**d**) Detail over the study site H1. (**e**) Detail over the study site H2. The hiking trails H1 and H2 are represented by red and blue lines.

**Figure 2 ijerph-16-04915-f002:**
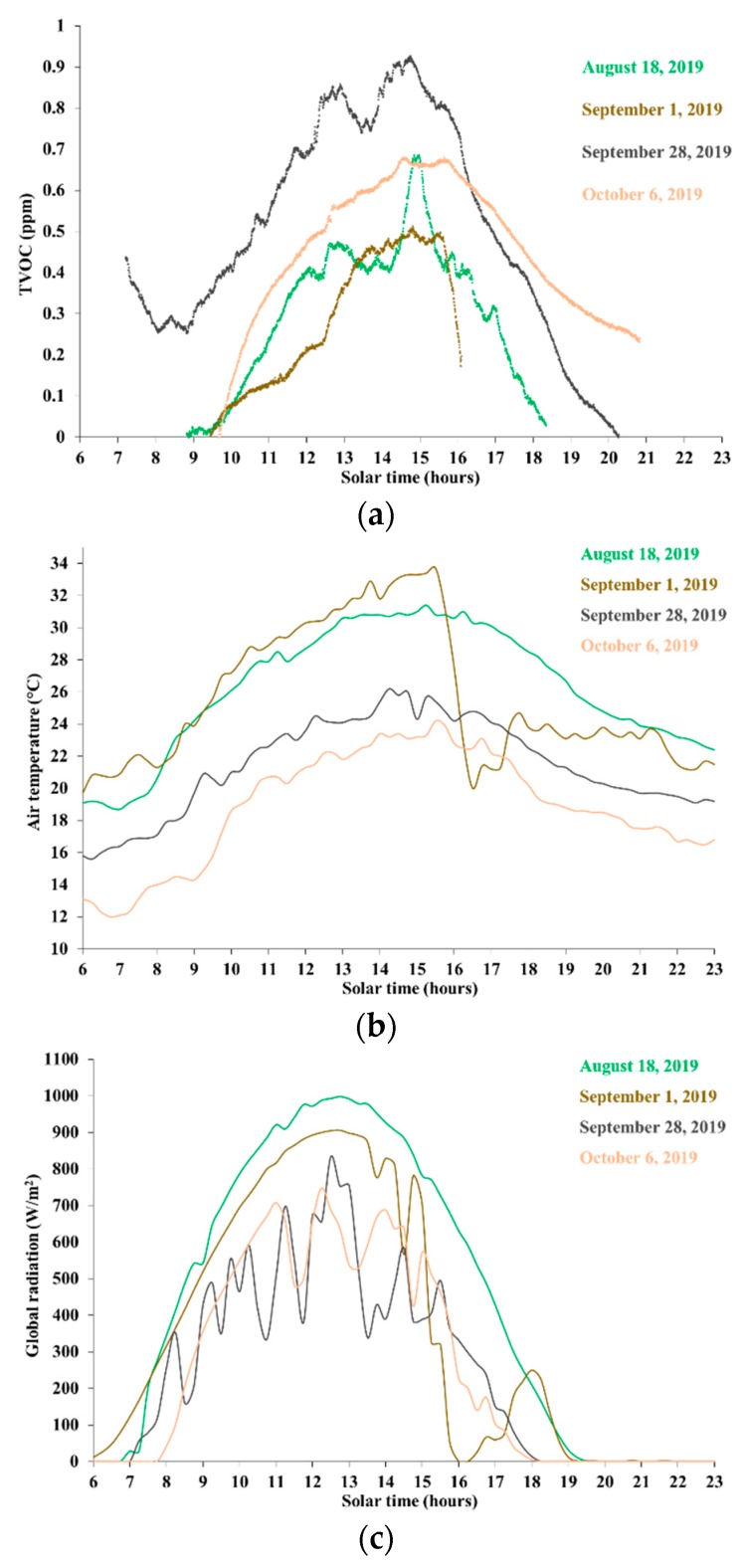
Data on site S1 during generally clear and calm days. (**a**) Total volatile organic compound (TVOC) concentration; (**b**) Air temperature; (**c**) Global radiation.

**Figure 3 ijerph-16-04915-f003:**
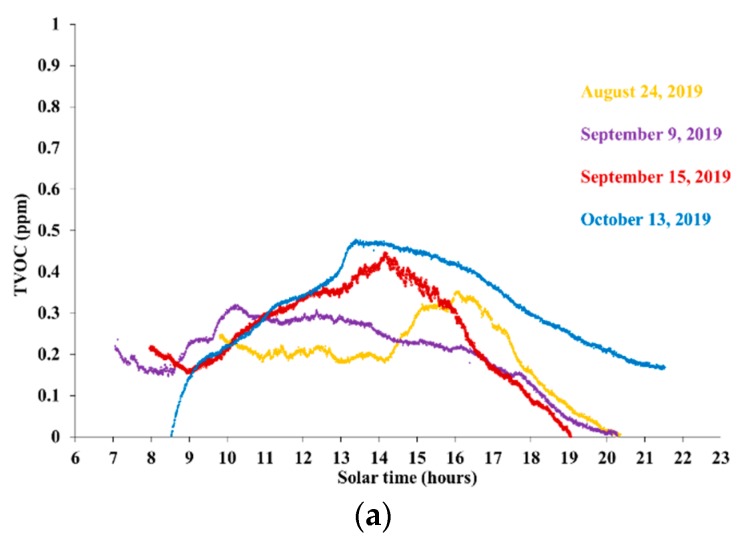
Data on site S1 during generally cloudy, colder, or windy days. (**a**) TVOC concentration; (**b**) Air temperature; (**c**) Global radiation; (**d**) Wind intensity.

**Figure 4 ijerph-16-04915-f004:**
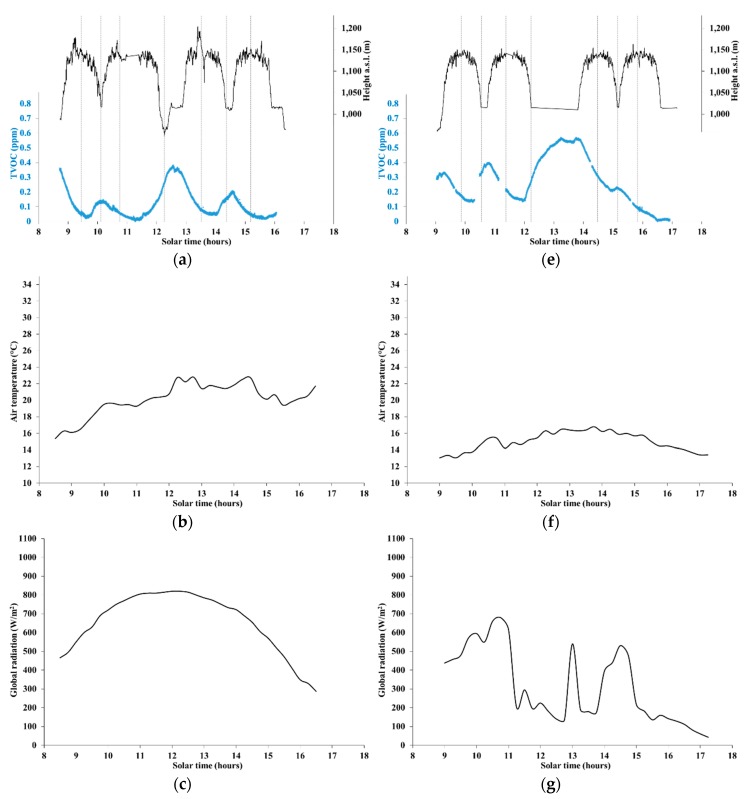
Data along forest path S2 on 11 September 2019. (**a**) Elevation and TVOC concentration; (**b**) Air temperature; (**c**) Global radiation; (**d**) Wind intensity. On 5 October 2019: (**e**) Elevation and TVOC concentration; (**f**) Air temperature; (**g**) Global radiation; (**h**) Wind intensity.

**Figure 5 ijerph-16-04915-f005:**
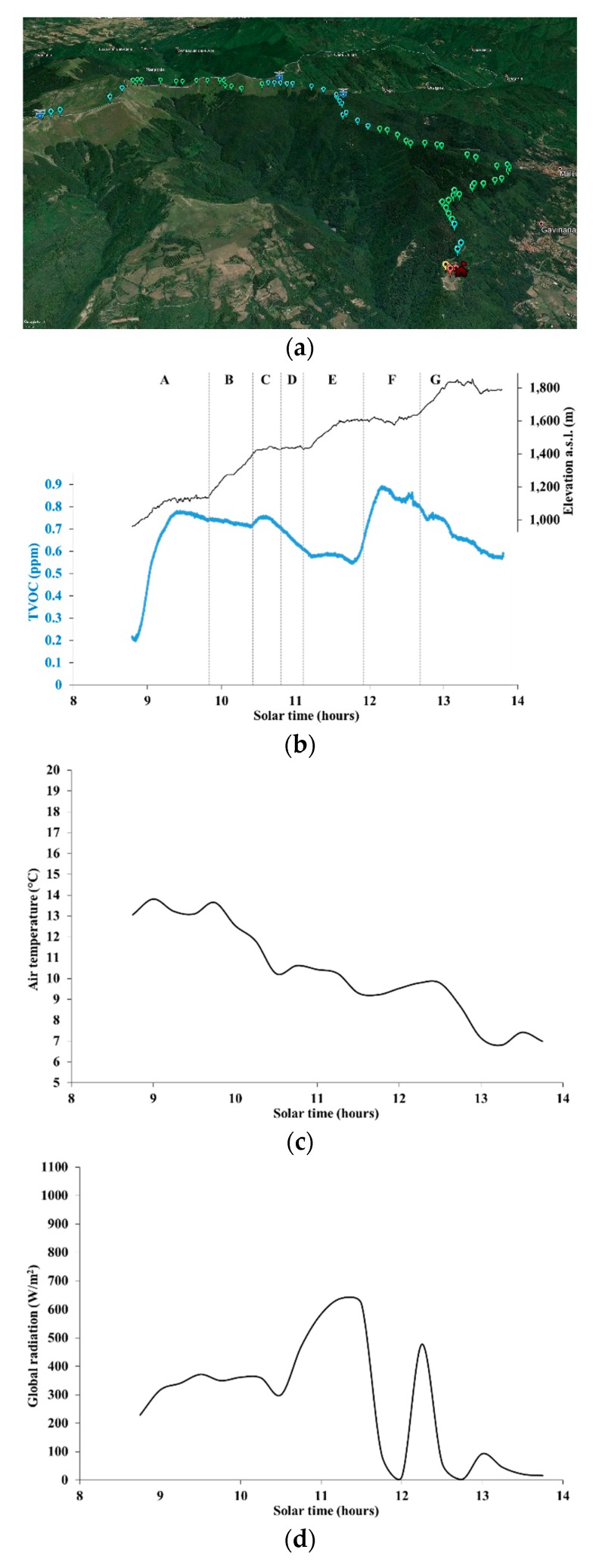
Data along the uphill part of the hiking trail H1 on 12 October 2019. (**a**) TVOC concentration classes on a topographical map (red, 0–0.225 ppm; yellow, 0.225–0.450 ppm; blue, 0.450–0.675 ppm; green, 0.675–0.892 ppm), with the red colored house representing the site “Pian dei Termini”, and the symbols with blue shield surmounted by an eagle representing mountain refuges, managed by the Italian Alpine Club; (**b**) Elevation and TVOC concentration; (**c**) Air temperature; (**d**) Global radiation; (**e**) Wind intensity.

**Figure 6 ijerph-16-04915-f006:**
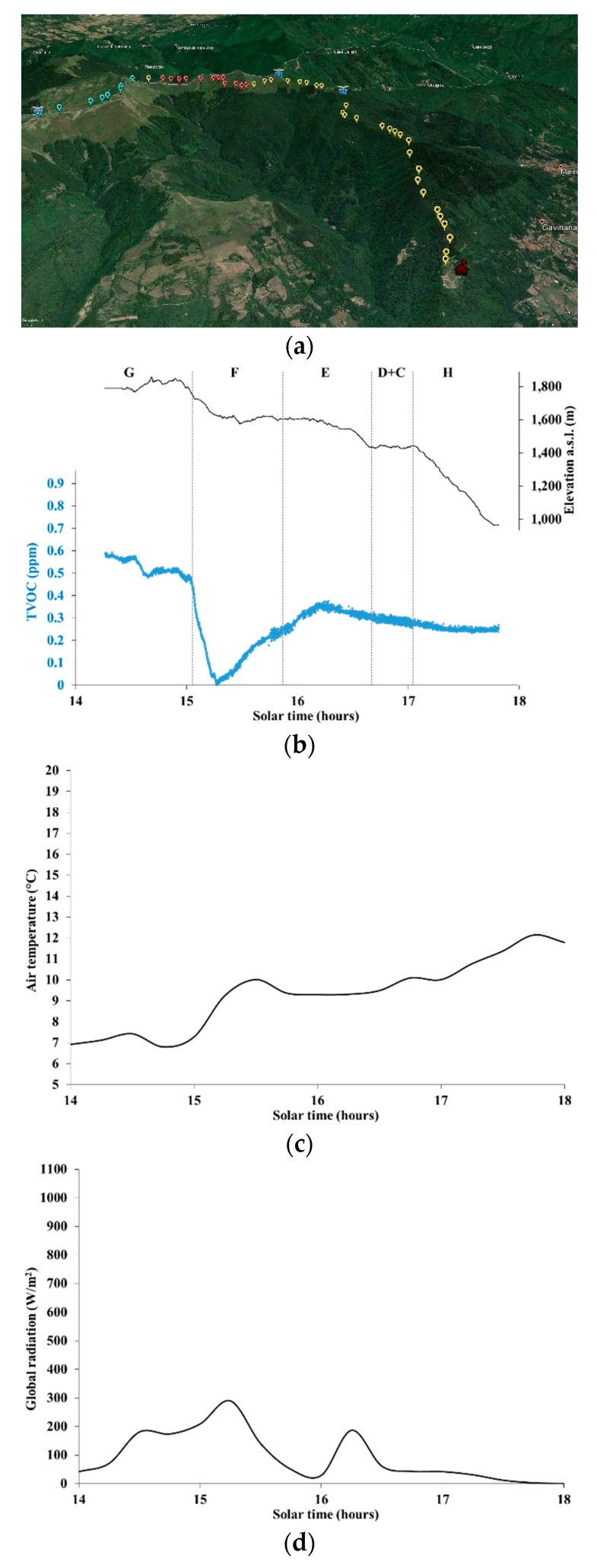
Data along the downhill part of the hiking trail H1 on 12 October 2019. (**a**) TVOC concentration classes on a topographical map (red, 0–0.225 ppm; yellow, 0.225–0.450 ppm; blue, 0.450–0.675 ppm; green, 0.675–0.892 ppm), with the red colored house representing the site “Pian dei Termini”, and the symbols with blue shield surmounted by an eagle representing mountain refuges, managed by the Italian Alpine Club; (**b**) Elevation and TVOC concentration; (**c**) Air temperature; (**d**) Global radiation; (**e**) Wind intensity.

**Figure 7 ijerph-16-04915-f007:**
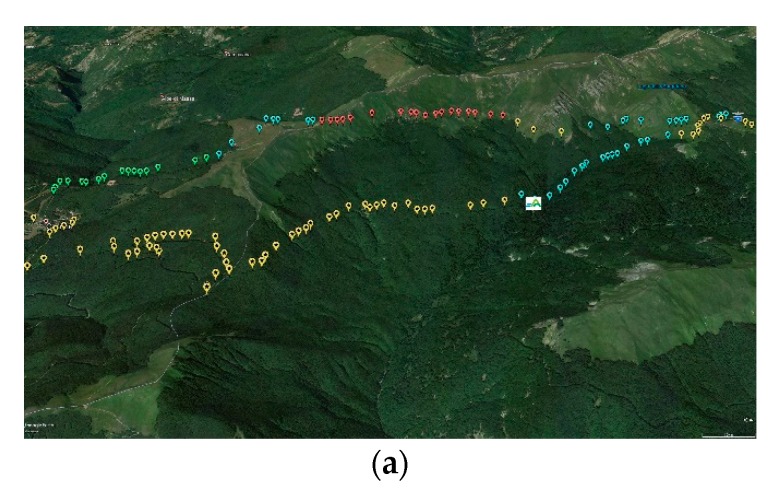
Data along the hiking trail H2 on 29 September 2019. (**a**) TVOC concentration classes on a topographical map (red, 0–0.194 ppm; yellow, 0.194–0.388 ppm; blue, 0.388–0.582 ppm; green, 0.582–0.776 ppm), with the square symbol with white background representing the “Segheria” refuge, managed by the Tuscan-Emilian Apennine National Park, and the symbol with blue shield surmounted by an eagle representing the “C. Battisti” refuge, managed by the Italian Alpine Club; (**b**) Elevation and TVOC concentration; (**c**) Air temperature; (**d**) Global radiation; (**e**) Wind intensity.

**Figure 8 ijerph-16-04915-f008:**
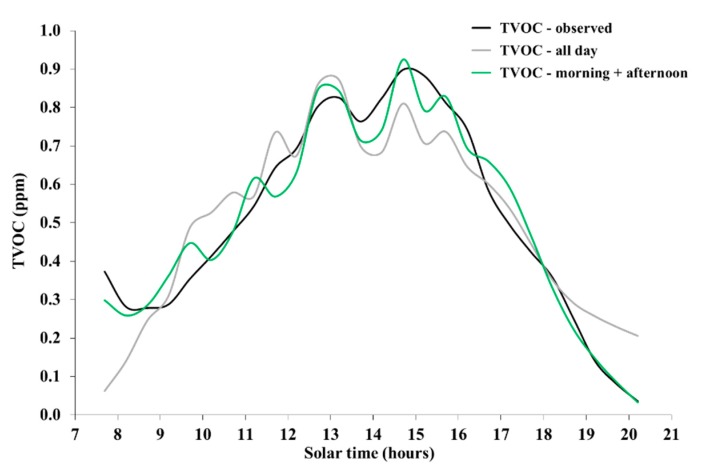
Observed and modeled concentration of volatile organic compounds on site S1 on 28 September 2019. The black line (TVOC—observed) represents the observed concentration. The gray line (TVOC—all day) represents the result of multiple regression over the whole period of measurements. The green line (TVOC—morning + afternoon) represents the result with regressions performed separately for morning and afternoon.

**Figure 9 ijerph-16-04915-f009:**
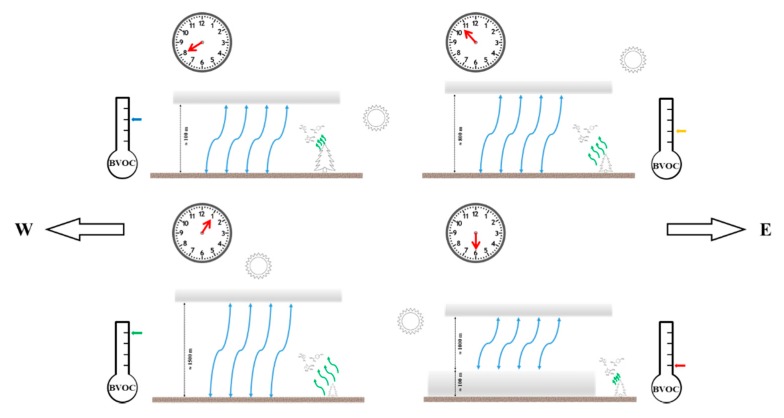
Conceptual model of biogenic volatile organic compound (BVOC) relative concentration at different times during daylight hours, on clear and calm days. The horizontal bars in shades of gray represent stable atmospheric layers. The curved lines in blue represent mixed layers. Curved lines in green represent BVOC emissions from trees and soil, with lengths indicatives of the respective emission rates. The east (E) and west (W) are indicated. Elements of the picture are not to scale.

**Table 1 ijerph-16-04915-t001:** List of study sites and weather stations.

ID	Type	Lat	Lon	Elevation	Data
S1 ^1^	Hilly site	43°41′21.16″	10°50′08.38″	61 m a.s.l.	TVOC
W1	Weather station	43°40′48.00″ N	10°53′00.40″ E	117 m a.s.l.	W, GR ^4^
W2	Weather station	43°40′48.00″ N	10°49′00.80″ E	102 m a.s.l.	T
S2 ^2^	Forest path			960–1150 m a.s.l.	TVOC
H1 ^2^	Hiking trail			960–1855 m a.s.l.	TVOC
W3	Weather station	44°01′12.00″ N	11°01′00.80″ E	950 m a.s.l.	GR
W4	Weather station	44°02′24.00″ N	10°55′00.20″ E	785 m a.s.l.	W
W5	Weather station	44°07′48.00″ N	10°46′00.80″ E	1716 m a.s.l.	T, W, RH
W6	Weather station	44°07′48.00″ N	10°44′00.40″ E	1000 m a.s.l.	T
H2 ^3^	Hiking trail			1312–1816 m a.s.l.	TVOC
W7	Weather station	44°12′00.00″ N	10°29′00.40″ E	1637 m a.s.l.	T, W, GR
W8	Weather station	44°12′36.00″ N	10°27′00.00″ E	1297 m a.s.l.	T
W9	Weather station	44°16′55.56″ N	10°24′00.10″ E	2057 m a.s.l.	T, W

^1^ Meteorological data from weather stations W1 and W2. ^2^ Meteorological data from weather stations W3, W4, W5, and W6. ^3^ Meteorological data from weather stations W7, W8, and W9. ^4^ Meteorological data abbreviations and units: T, air temperature 2 m above ground (°C); W, wind intensity 10 m above ground (ms^−1^); RH, relative humidity 2 m above ground (%); GR, global radiation (Wm^−2^); TVOC, total volatile organic compound (mg/kg).

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
