# Peer review of "Temporal and Spatial Variability of Volatile Organic Compounds in the Forest Atmosphere"

_ijerph, 2019, doi:10.3390/ijerph16244915_

Round 1
Reviewer 1 Report
This study addresses some aspects of the concentration of volatile organic compounds in forest areas in Italy. The aims and objectives of the paper fit in the editorial policy of this journal. I think the authors have made a worthy effort, and the manuscript is promising. However, my overall recommendation is that the manuscript should be not accepted in its present form. My comments are listed as follows:
General comment:
• Under my view, the main weakness of this manuscript is the lack of sound explanations between diverse parameters measured. These relations have been not shown with clarity, and the reader misses some statistical analysis among some of the variables analysed. If the authors do not provide that results, a clear justification should be included in the paper (it is not enough l. 507-508).
Main comments:
• l. 363: the hypothesis should be introduced in the Methodology Section, not in the Discussion Section
• Discussion Section: all the future research lines should be aggregated in the same paragraph, not disseminated along this Section.
• Conclusions Section could be partially rewritten (e.g., delete the sentence in l. 571-573, and the last paragraph fits better in Discussion Section).
Minor comments:
• l. 71: maybe there is a typo.
Author Response
Response to Reviewer 1 Comments
This study addresses some aspects of the concentration of volatile organic compounds in forest areas in Italy. The aims and objectives of the paper fit in the editorial policy of this journal. I think the authors have made a worthy effort, and the manuscript is promising. However, my overall recommendation is that the manuscript should be not accepted in its present form. My comments are listed as follows:
Response: The authors gratefully acknowledge the esteemed Reviewer for her/his insightful comments.
General comment:
Point 1: Under my view, the main weakness of this manuscript is the lack of sound explanations between diverse parameters measured. These relations have been not shown with clarity, and the reader misses some statistical analysis among some of the variables analysed. If the authors do not provide that results, a clear justification should be included in the paper (it is not enough l. 507-508).
Response 1: Thank you very much for this comment – it goes straight to the point. The fourth paragraph of the revised version of the manuscript expands the explanation:
“Building a comprehensive quantitative model for BVOC concentration as a function of time of the day and meteorological data, even on a single site and for clear and calm days, would require much more extensive measurements. As well, the seasonality of emission rates from plants and soil should be included, in turn depending on the specific forest composition. The structure and the hourly evolution of the atmospheric boundary layer, including its vertical stratification, should be accurately represented, too. The climatology of BVOC concentrations along a path would be even more challenging, due to changing forest composition and exposure of the path. On the other hand, mapping BVOC concentration near the ground is just in its infancy, and the road ahead is still very long.”
The point is that this was the first time, to our knowledge, that such high-frequency measurements were made, and we had few clues in advance about the output. Now we know that continuous measurement campaigns should be performed, in order to derive meaningful relationships, taking into account all the forcing variables.
However, based on your insightful comment, we performed a preliminary modeling of the concentration data collected during a single day, on a single site, described in Figure 8 and the text above and below it, including Equation (1) and Equation (2). We found preliminary confirmation to our hypotheses: solar radiation plays substantially opposite roles in the morning and in the afternoon, with regard to near ground BVOC concentration, which is likely related to the complex evolution of the atmospheric boundary layer. Further explanation is available in the text below the equations. This is a promising result, but the road ahead towards a comprehensive model is still very long (and very exciting).
Main comments:
Point 2: l. 363: the hypothesis should be introduced in the Methodology Section, not in the Discussion Section.
Response 2: This hypothesis has been introduced at the end of Section 2.3 (Materials and methods), recalled few times in Section 3 (Results), and elaborated in Section 4 (Discussion).
Point 3: Discussion Section: all the future research lines should be aggregated in the same paragraph, not disseminated along this Section.
Response 3: Thank you for this comment. All the future research lines have been aggregated in a list towards the end of Section 4 (Discussion).
Point 4: Conclusions Section could be partially rewritten (e.g., delete the sentence in l. 571-573, and the last paragraph fits better in Discussion Section).
Response 4: About the sentence in lines 571-573, the other esteemed Reviewer suggested introducing further guidelines right for hikers in Section 4 (Discussion), what we have done, thus we have left this sentence. About the last paragraph, your comment is completely right and it has been moved to the end of Section 4 (Discussion), and marginally shortened. Other minor changes were made to Section 5 (Conclusions), to account for the introduced changes to the manuscript, and for the sake of readability.
Minor comments:
Point 5: l. 71: maybe there is a typo.
Response 5: Section 1 (Introduction) has been deeply reworked and the text has been checked – we hope to have fixed most of errors including typos.

Reviewer 2 Report
The introduction is a little long, authors could think cut some "irrelevant" sentences. for example, Line 61-64, since this study is only focusing on aromatic volatile substances, not need to mention "sound of stream water", "landscape", etc.. Line 74-77, too many details about the positive effects of forest therapy since this paper is not describing the immune system. Overall, authors should try to shorten the Introduction, with precise and short sentences. Right now it reads more like newspaper.
For Figure 1, probably make the background more transparent, so readers can easily see the path. Right now it is hard to see red and blue lines.
For analyzing the data, I wonder if authors could perform some more sophisticated statistical approaches instead of just looking at the patterns by eyes for each path/site at different dates. Multiple linear regressions could be used to test impacts of temperature, radiation, wind and their interactions on TVOC concentrations; ANOVA can be used to test the differences of TVOC concentrations between different dates or at different sites;
And if authors can build a strong relationship between TVOC concentrations and these independent factors, authors could estimate the total emitted concentrations throughout the year for each trial/site based on the monitored meteorological data from the weather station. And correlated that with the satisfaction rating for each trial from the tourists (can use number of tourists throughout the year to indicate if there is?). The goal is to possible see if the TVOC is important for appealing tourists and if there is some
And right now this paper is too long, some figures could be merged together, for example, Figure 4 and 5 could be combined together since they are describing the same site for different dates. And for all figures, authors should make both x and y axis lables larger.
Line 474-505, these conclusions are important, authors could take a step further to make suggestions on hiking at each trial based on the their BVOC emissions and hiking miles. for example, suggesting tourists to start hike which trial at noon instead of morning, something like that? This extra discussion can help expand the implications of this study.
Probably replace the "BVOC" with "TVOC" in the Discussion section since this study is reporting data about TVOC. For example, line 458-line 460, readers would wonder is BVOC and TVOC the same thing? why not just use one to use TBVOC?
Author Response
Response to Reviewer 2 Comments
The authors gratefully acknowledge the esteemed Reviewer for her/his insightful comments.
Point 1: The introduction is a little long, authors could think cut some "irrelevant" sentences. for example, Line 61-64, since this study is only focusing on aromatic volatile substances, not need to mention "sound of stream water", "landscape", etc.. Line 74-77, too many details about the positive effects of forest therapy since this paper is not describing the immune system. Overall, authors should try to shorten the Introduction, with precise and short sentences. Right now it reads more like newspaper.
Response 1: Thank you very much for this comment, which helped streamline the manuscript. The Introduction section has been remarkably shortened, removing unnecessary sentences and paragraphs.
Point 2: For Figure 1, probably make the background more transparent, so readers can easily see the path. Right now it is hard to see red and blue lines.
Response 2: Thank you very much for this comment, details – especially in Figure 1(a) – were barely readable. Figure 1 has been reworked, and two other panels have been added, to make details more readable. The panels have also been enlarged (they can be easily resized, based on editorial decision). Now, each study site is represented in greater detail, and all the paths (forest path and hiking trails) appear to be readable.
Point 3: For analyzing the data, I wonder if authors could perform some more sophisticated statistical approaches instead of just looking at the patterns by eyes for each path/site at different dates. Multiple linear regressions could be used to test impacts of temperature, radiation, wind and their interactions on TVOC concentrations; ANOVA can be used to test the differences of TVOC concentrations between different dates or at different sites.
Response 3: Thank you very much for this comment – it goes straight to the point. The fourth paragraph of the revised version of the manuscript provides some explanation:
“Building a comprehensive quantitative model for BVOC concentration as a function of time of the day and meteorological data, even on a single site and for clear and calm days, would require much more extensive measurements. As well, the seasonality of emission rates from plants and soil should be included, in turn depending on the specific forest composition. The structure and the hourly evolution of the atmospheric boundary layer, including its vertical stratification, should be accurately represented, too. The climatology of BVOC concentrations along a path would be even more challenging, due to changing forest composition and exposure of the path. On the other hand, mapping BVOC concentration near the ground is just in its infancy, and the road ahead is still very long.”
The point is that this was the first time, to our knowledge, that such high-frequency measurements were made, and we had few clues in advance about the output. Now we know that continuous measurement campaigns should be performed, in order to derive meaningful relationships, taking into account all the forcing variables. We hope that this study can pave the way towards substantial further research in such a new and important field.
However, based on your insightful comment, we performed a preliminary modeling of the concentration data collected during a single day, on a single site, described in Figure 8 and the text above and below it, including Equation (1) and Equation (2). We found preliminary confirmation to our hypotheses: solar radiation plays substantially opposite roles in the morning and in the afternoon, with regard to near ground BVOC concentration, which is likely related to the complex evolution of the atmospheric boundary layer. Further explanation is available in the text below the equations.
We tried as well to compare data on a single site, at different dates, but linear regression showed ineffective. There are several reasons for that, among which the following ones. Measurements were taken at different times in a season (late summer to early fall) with rapidly declining solar radiation, leading to different evolution and properties of the atmospheric boundary layer; as well, it is known that most plants show rapidly declining emissions from summer to fall, irrespective of the daily temperatures. We think that these reasons are sufficient to explain the difficulty to derive comprehensive quantitative relationships with the available data.
The presented results, including the newly introduced, simple regression model, are quite promising, but the road ahead towards a comprehensive model is still very long (and very exciting).
Point 4: And if authors can build a strong relationship between TVOC concentrations and these independent factors, authors could estimate the total emitted concentrations throughout the year for each trial/site based on the monitored meteorological data from the weather station. And correlated that with the satisfaction rating for each trial from the tourists (can use number of tourists throughout the year to indicate if there is?). The goal is to possible see if the TVOC is important for appealing tourists and if there is some.
Response 4: This is an invaluable suggestion! Thank you so much. While our previous response to Point 3 explains why your direction is unfeasible right now, based on the available data, yours is nothing less than an insightful and perspective research plan. We warmly hope that ourselves, or other scholars, will be able to go ahead along this direction, which will bring considerable benefits to everybody.
Point 5: And right now this paper is too long, some figures could be merged together, for example, Figure 4 and 5 could be combined together since they are describing the same site for different dates. And for all figures, authors should make both x and y axis labels larger.
Response 5: The text of the manuscript has been shortened, particularly Section 1 (Introduction), based on the comments produced by the esteemed Reviewers. As well, Figure 4 and 5 have been combined together, thus all subsequent Figures have been renumbered (in the text, too).
All charts have been reworked, enlarging the labels on both x and y-axes, as well as any other label present in such charts.
Point 6: Line 474-505, these conclusions are important, authors could take a step further to make suggestions on hiking at each trial based on the their BVOC emissions and hiking miles. for example, suggesting tourists to start hike which trial at noon instead of morning, something like that? This extra discussion can help expand the implications of this study.
Response 6: Thank you, this is really a very insightful comment. We have added another conclusion in Section 4 (“Discussion”), reading as follows:
“With regard to BVOCs inhalation, during clear and calm days, the preliminary planning suggestion for hikers can be issued, that forest trails should be preferably traveled in early morning, and noon to early afternoon, rather than in mid-morning and late afternoon. Depending on hiking miles, the latter periods could be used to rest at mountain refuges, or to walk through areas devoid of forest vegetation.”
Yet another conclusion was added, namely:
“Cloudy or windy days appear less suitable at any time, to enjoy the BVOCs-related forest healing effects, due to generally lower and largely unpredictable concentration levels, adding this flaw to the generally lower level of comfort.”
Point 7: Probably replace the "BVOC" with "TVOC" in the Discussion section since this study is reporting data about TVOC. For example, line 458-line 460, readers would wonder is BVOC and TVOC the same thing? why not just use one to use TBVOC?
Response 7: Thank you for this comment, we acknowledge that some confusion could be generated. A paragraph from moved from the latter part of Section 4 (Discussion) to the first part, and marginally changed, in order to clarify:
“The meaningfulness of this study and the related findings relies upon the representativeness of TVOC concentrations measured by means of a PID with respect to the biogenic components (BVOC). Since the components of the detected TVOCs were not analyzed in any study sites and paths, in principle their origin was unknown, i.e., they could be either biogenic, or anthropogenic, or both. However, the remoteness of the areas into consideration from anthropogenic sources, as well as the adoption of a variable offset concentration level, equal to the absolute minimum during any measurement day, with respect to which all the other data were referred, should have minimized the chance of substantial contributions of anthropogenic sources to the reported TVOC concentration data.”.
As well, a statement was added at the end of the above paragraph:
“Based on the above, in the following, unless explicitly specified, the volatile organic compounds under consideration will be referred to as BVOCs.”
Moreover, we have use abbreviations more sparingly in Section 4 (Discussion), again to avoid confusing readers.
